# SAMPLING FROM ENERGY-BASED POLICIES USING DIFFUSION

## ABSTRACT

Energy-based policies offer a flexible framework for modeling complex, multimodal behaviors in reinforcement learning (RL). In maximum entropy RL, the optimal policy is a Boltzmann distribution derived from the soft Q-function, but direct sampling from this distribution in continuous action spaces is computationally intractable. As a result, existing methods typically use simpler parametric distributions, like Gaussians, for policy representation — limiting their ability to capture the full complexity of multimodal action distributions. In this paper, we introduce a diffusion-based approach for sampling from energy-based policies, where the negative Q-function defines the energy function. Based on this approach, we propose an actor-critic method called Diffusion Q-Sampling (DQS) that enables more expressive policy representations, allowing stable learning in diverse environments. We show that our approach enhances exploration and captures multimodal behavior in continuous control tasks, addressing key limitations of existing methods.

## 1 INTRODUCTION

Deep reinforcement learning (RL) is a powerful paradigm for learning complex behaviors in diverse domains, from strategy-oriented games (Silver et al., 2016; Berner et al., 2019; Schrittwieser et al., 2020) to fine-grained control in robotics (Kober et al., 2013; Sünderhauf et al., 2018; Wu et al., 2023). In the RL framework, an agent learns to make decisions by interacting with an environment and receiving feedback in the form of reward. The agent aims to learn a policy that maximizes the cumulative sum of rewards over time by exploring actions and exploiting known information about the environment's dynamics.

The parameterization of the policy is a crucial design choice for any RL algorithm. Under the conventional notion of optimality, under full observability, there always exists an optimal deterministic policy that maximizes the long-term (discounted) return (Sutton & Barto, 2018). However, this is only true when the agent has explored sufficiently and has nothing to learn about the environment. Exploration requires a stochastic policy, to experiment with different potentially rewarding actions. Moreover, even in the exploitation phase, there may be more than one way of performing a task and we might be interested in mastering all of them. This diversification is motivated by the robustness of the resulting stochastic policy to environment changes; if certain pathways for achieving a task become infeasible due to a change of the dynamics or reward, some may remain feasible, and the agent has an easier time in adapting to this change by exploiting and improving the viable options. This argument also suggests that such policies can also serve as effective initialization for fine-tuning on specific tasks.

While exploration, diversity and robustness motivate stochastic policies, representing such policies in continuous action spaces remains challenging. As a result, stochasticity is often introduced by noise injection (Lillicrap et al., 2015) or using an arbitrary parametric family (Schulman et al., 2015) which lacks expressivity.

Orthogonal to the difficulty of representing such policies is their training objective; policies are often optimized to maximize the Q-function, and stochasticity is introduced to encourage exploration as an afterthought. However, our argument for stochasticity favours multi-modal policies; instead of learning the *single best* way to solve a task, we want to learn *all reasonably good* ways to solve the task.

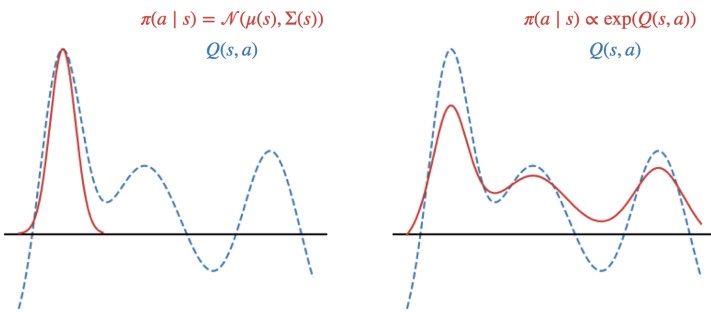

Figure 1: Illustration comparing Gaussian and Boltzmann policies.

We address both of these issues using diffusion-based sampling from a Boltzmann policy. Diffusion models offer a potential solution to the policy parameterization problem since they are expressive and can produce high-quality samples from complex distributions. Indeed, they have been extensively applied to solve sequential decision-making tasks, especially in offline settings where they can model multimodal datasets from suboptimal policies or diverse human demonstrations. A few studies have applied these models in the online setting, focusing on deriving training objectives for policy optimization via diffusion. Yang et al. (2023); Psenka et al. (2023) use the gradient of the Q-function to guide action sampling, however, the exact form of the policy is unspecified and it is unknown what distribution the diffusion models sample from.

We consider the problem of explicitly sampling from energy-based policies of the form,

$$\pi(a \mid s) \propto \exp(Q^\pi(s, a)).$$

This is also known as the Boltzmann distribution of the Q-function. The optimal policy in the maximum entropy RL framework is also known to be of this form, except it uses the soft Q-function (Haarnoja et al., 2017). Such a policy has several benefits. First, it offers a principled way to balance exploration and exploitation for continuous action spaces. By sampling from this distribution, the policy still prioritizes actions with high Q-values but also has a non-zero probability of sampling sub-optimal actions. While the use of Boltzmann policies is common in the discrete setting, it is challenging in continuous spaces; this sampling problem is often tackled with Markov Chain Monte Carlo (MCMC) techniques, which can be computationally expensive and suffer from exponential mixing time. Second, this formulation naturally incorporates multimodal behavior, since the policy can sample one of multiple viable actions at any given state.

Our contributions in this work are as follows:

- We propose a novel framework for sequential decision-making using diffusion models for sampling from energy-based policies.
- We propose an actor-critic algorithm for training diffusion policies based on our framework.
- We demonstrate the effectiveness of this approach for learning multimodal behaviors in maze navigation tasks.
- We demonstrate improved sample efficiency in continuous control tasks owing to better exploration capabilities of our method.

## 2 RELATED WORK

Our work is related to two distinct sub-areas of reinforcement learning: the relatively new and actively explored line of work on applying diffusion models in the RL setting, and the classical maximum entropy RL framework.

**Diffusion models in RL.** Early work on applying diffusion models for RL was focused on behavior cloning in the offline setting (Chi et al., 2023). This setting more closely matches the original purpose of diffusion models - to match a distribution to a given dataset. Janner et al. (2022); Ajay et al. use a diffusion model trained on the offline data for trajectory optimization, while Reuss et al.

(2023); Jain & Ravanbakhsh (2024) apply diffusion models to offline goal-reaching tasks by additionally conditioning the score function on the goal. Within the behavior cloning setting, there is some existing work on learning a stochastic state dynamics model using diffusion (Li et al., 2022).

Beyond behavior cloning, offline RL methods incorporate elements from Q-learning to learn a value function from the offline dataset and leverage the learned Q-function to improve the diffusion policy. A large body of work exists in this sub-field, where the most common approach is to parameterize the policy using a diffusion model and propose different training objectives to train the diffusion policy. Wang et al.; Kang et al. (2024) add a Q-function maximizing term to the diffusion training objective, and Hansen-Estruch et al. (2023) use an actor-critic framework based on a diffusion policy and Implicit Q-learning (Kostrikov et al., 2021). Lu et al. (2023) take an energy-guidance approach, where they frame the problem as using the Q-function to guide the behavior cloning policy to high reward trajectories.

The application of diffusion models has been relatively less explored in the online setting (Ding & Jin, 2023). Yang et al. (2023) modifies actions in the replay buffer based on the gradient of the Q-function, then trains a diffusion model on the modified actions. Psenka et al. (2023) directly trains a neural network to match the gradient of the Q-function, then uses it for sampling. The policy resulting from these methods does not have an explicit form, and it is unknown what exact distribution these diffusion models sample from. In contrast, our method explicitly models the policy as a Boltzmann distribution and samples actions from this distribution.

**Maximum entropy RL.** In contrast to standard RL, where the goal is to maximize expected returns, in the maximum entropy RL framework, the value function is augmented by Shannon entropy of the policy. Ziebart et al. (2008) applied such an approach in the context of inverse reinforcement learning and Haarnoja et al. (2017) generalized this approach by presenting soft Q-learning to learn energy-based policies. A follow-up work, Haarnoja et al. (2018a), presented the well-known soft actor-critic (SAC) algorithm. This line of work proposes to learn a soft value function by adding the entropy of the policy to the reward. The optimal policy within this framework is a Boltzmann distribution, where actions are sampled based on the exponentiated soft Q-values.

A separate but related line of work on generative flow networks (GFlowNets), originally defined in the discrete case (Bengio et al., 2021; 2023), learns a policy that samples terminal states proportional to the Boltzmann density corresponding to some energy function. They have been extended to the continuous setting (Lahlou et al., 2023) and under certain assumptions, they are equivalent to maximum entropy RL (Tiapkin et al., 2024; Deleu et al.). They can effectively sample from the target distribution using off-policy exploration, however, they encounter challenges in credit assignment and exploration efficiency (Malkin et al., 2022; Madan et al., 2023; Rector-Brooks et al., 2023; Shen et al., 2023). Our approach is distinct as we sample the action at each step from the Boltzmann density of the Q-function, instead of the terminal states based on the reward.

## 3 PRELIMINARIES

### 3.1 REINFORCEMENT LEARNING

We consider an infinite-horizon Markov Decision Process (MDP) denoted by the tuple $(\mathcal{S}, \mathcal{A}, \mathcal{P}, r, \gamma)$, where the state space $\mathcal{S}$ and action space $\mathcal{A}$ are continuous. $\mathcal{P} : \mathcal{S} \times \mathcal{A} \times \mathcal{S} \rightarrow [0, 1]$ denotes the transition probability of the next state $\mathbf{s}_{t+1} \in \mathcal{S}$ given the current state $\mathbf{s}_t \in \mathcal{S}$ and action $\mathbf{a}_t \in \mathcal{A}$. The reward function $r : \mathcal{S} \times \mathcal{A} \rightarrow \mathbb{R}$ is assumed to be bounded $r(\mathbf{s}, \mathbf{a}) \in [r_{\min}, r_{\max}]$. $\gamma \in (0, 1]$ is the discount factor.

A policy $\pi : \mathcal{S} \rightarrow \mathcal{A}$ produces an action for every state $s \in \mathcal{S}$. In the standard RL framework, the objective is to learn a policy that maximizes the expected sum of rewards $\sum_t \mathbb{E}_{a_t \sim \pi(\cdot|s_t), s_{t+1} \sim \mathcal{P}(\cdot|s_t, a_t)} [r(s_t, a_t)]$.

Actor-critic is a commonly used framework to learn such policies. It consists of optimizing a policy (the actor) to choose actions that maximize the action value function, also called the Q-function (the critic). The Q-function is defined as the sum of expected future rewards starting from a given

state-action pair, and thereafter following some policy $\pi$,

$$Q^\pi(s, a) = \mathbb{E}_\pi \left[ \sum_{k=t}^\infty r(s_k, a_k) \mid s_t = s, a_t = a \right].$$

The optimal policy is defined as the policy that maximizes the sum of rewards along a trajectory,

$$\pi^* = \arg \max_\pi \mathbb{E}_\pi \left[ \sum_{t=0}^\infty r(s_t, a_t) \right]$$

## 3.2 DIFFUSION MODELS

**Denoising Diffusion.** Denoising diffusion (Dinh et al., 2016; Ho et al., 2020; Song et al., 2021) refers to a class of generative models which relies on a stochastic process which progressively transforms the target data distribution to a Gaussian distribution. The time-reversal of this diffusion process gives the generative process which can be used to transform noise into samples from the target data distribution.

The forward noising process is a stochastic differential equation,

$$dx_\tau = -\alpha(\tau)x_\tau d\tau + g(\tau)dw_\tau \tag{1}$$

where $w_\tau$ denotes Brownian motion. In this paper, we consider the Variance Exploding (VE) SDE where the decay rate, $\alpha$, is set to $\alpha(\tau) = 0$. This noising process starts with samples from the target density $x_0 \sim p_0$ and progressively adds noise to them over a diffusion time interval $\tau \in [0, 1]$. The marginal probability distribution at time $\tau$ is denoted by $p_\tau$ and is the convolution of the target density $p_0$ with a normal distribution with a time-dependent variance, $\sigma_\tau^2$. For the VE setting we consider, these marginal distributions are given by,

$$p_\tau(x_\tau) = \int_0^\tau p_0(x_0)\mathcal{N}(x_\tau; x_0, \sigma_\tau^2)dx_0 \tag{2}$$

where the variance is related to the diffusion coefficient, $g(\tau)$ via $\sigma_\tau^2 = \int_0^\tau g(\xi)^2 d\xi$.

The generative process corresponding to the corresponding to 1 is an SDE with Brownian motion $\bar{w}_\tau$, given by:

$$dx_\tau = [-\alpha(\tau)x_\tau - g(\tau)^2 \nabla \log p_\tau(x_\tau)]d\tau + g(\tau)d\bar{w}_\tau \tag{3}$$

Therefore, to be able to generate data, we need to estimate the score of the intermediate distributions, $\nabla \log p_\tau(x_\tau)$.

**Iterated Denoising Energy Matching.** Recently, Akhound-Sadegh et al. (2024) proposed an algorithm known as iDEM (Iterated Denoising Energy Matching) for sampling from a Boltzmann-type target distribution, $p_0(x) \propto \exp(-\mathcal{E}(x))$, where $\mathcal{E}$ denotes the energy. iDEM is a diffusion-based neural sampler, which estimates the diffusion score, $\nabla \log p_\tau$ using a Monte Carlo estimator. Given the VE diffusion path defined above, iDEM rewrites the score of the marginal densities as,

$$\begin{aligned} \nabla \log p_\tau &= \frac{\int \nabla \exp(-\mathcal{E}(x_0))\mathcal{N}(x_\tau; x_0, \sigma_\tau^2)dx_0}{\int \exp(-\mathcal{E}(x_0))\mathcal{N}(x_\tau; x_0, \sigma_\tau^2)dx_0} \\ &= \frac{\mathbb{E}_{\tilde{x}\sim\mathcal{N}(x_\tau, \sigma_\tau^2)}\left[\nabla \exp(-\mathcal{E}(x)\right]}{\mathbb{E}_{\tilde{x}\sim\mathcal{N}(x_\tau, \sigma_\tau^2)}\left[\exp(-\mathcal{E}(x)\right]} \end{aligned} \tag{4}$$

Rearranging the equation above leads to the $K$-sample Monte-Carlo estimator of the score, given by,

$$S_k(x_\tau, \tau) = \nabla \log \sum_{i=1}^K \exp\left(-\mathcal{E}(\tilde{x}^{(i)})\right), \quad \tilde{x}^{(i)} \sim \mathcal{N}(x_\tau, \sigma_\tau^2) \tag{5}$$

A score-network, $f_\phi$ is trained to regress to the MC estimator, $S_K$. The network is trained using a bi-level iterative scheme: (1) in the outer-loop a replay buffer is populated with samples that are generated using the model and (2) in the inner-loop the network is regressed $S_k(x_\tau, \tau)$ where $x_\tau$ are noised samples from the replay buffer.

## 4 METHOD

Our objective is to learn general policies of the form $\pi(\mathbf{a} \mid \mathbf{s}) \propto \exp(-\mathcal{E}(\mathbf{s}, \mathbf{a}))$, where $\mathcal{E}$ represents an energy function which specifies the desirability of state-action pairs. By setting the Q-function, $Q(\mathbf{s}, \mathbf{a})$ as the negative energy, we get what is known as the Boltzmann policy,

$$\pi(\mathbf{a} \mid \mathbf{s}; T) = \frac{\exp(\frac{1}{T}Q(\mathbf{s}, \mathbf{a}))}{\int_{\mathbf{a}} \exp(\frac{1}{T}Q(\mathbf{s}, \mathbf{a}))\mathrm{d}\mathbf{a}}. \tag{6}$$

Choosing such a policy gives us a principled way to balance exploration and exploitation. Specifically, by scaling the energy function with a temperature parameter $T$ and annealing it to zero, we get a policy that initially explores to collect more information about the environment and over time exploits the knowledge it has gained.

### 4.1 BOLTZMANN POLICY ITERATION

We propose a policy iteration process that alternates between learning the Q-function of the current policy and optimizing the policy to sample from the Boltzmann distribution of this Q-function. While Boltzmann policies have been used in Q-learning methods for discrete action spaces, our approach introduces a key difference – we rely on the Q-function of the current policy, rather than the optimal Q-function. Since the policy is not maximizing the Q-function but rather following a Boltzmann distribution, and the Q-function corresponds to the current policy, it is not immediately clear that this iterative process of policy evaluation and improvement will converge to a unique fixed point. Indeed, it is conceivable to encounter policy-value pairs that are quite sub-optimal yet consistent, where the policy is Boltzmann in $Q$, and $Q$ is the value function of that specific policy. The following results show that these updates still constitute policy iteration and there is a unique fixed point. Moreover, such a fixed point is bound to recover the optimal policy.

**Policy Evaluation.** In the policy evaluation step, we wish to estimate the value function associated with a policy $\pi$. For a fixed policy, the Q-function is computed iteratively, starting from any function $Q : \mathcal{S} \times \mathcal{A} \to \mathbb{R}$ and repeatedly applying the Bellman backup operator $\mathcal{T}^\pi$ given by,

$$\mathcal{T}^\pi Q(\mathbf{s}_t, \mathbf{a}_t) \triangleq r(\mathbf{s}_t, \mathbf{a}_t) + \gamma \mathbb{E}_{\mathbf{s}_{t+1}\sim\mathcal{P},\mathbf{a}_{t+1}\sim\pi}\left[Q(\mathbf{s}_{t+1}, \mathbf{a}_{t+1})\right] \tag{7}$$

This is the standard Bellman backup and the usual convergence results for policy evaluation can be applied (Sutton & Barto, 2018).

**Policy Improvement.** In the policy improvement step, we update the policy to match the Boltzmann density of the current Q-function. The following lemma shows that this choice of policy update guarantees that the new policy is better than the old policy in terms of the Q-values. Here, $\Pi$ denotes some set of policies, which corresponds in our case to the set of policies represented by the diffusion process.

**Lemma 1.** *Let $\pi_{old} \in \Pi$ and $\pi_{new}$ be the policy from Equation* (6) *with respect to $Q^{\pi_{old}}$. Then $Q^{\pi_{new}}(\mathbf{s}_t, \mathbf{a}_t) \geq Q^{\pi_{old}}(\mathbf{s}_t, \mathbf{a}_t)$ for all $\mathbf{s}_t, \mathbf{a}_t \in \mathcal{S}, \mathcal{A}$.*

*Proof.* See Appendix A.1 □

**Policy Iteration.** By alternating between policy evaluation and policy improvement, we can prove that it converges to the optimal policy within $\Pi$.

**Theorem 1.** *Alternating between policy evaluation and policy improvement repeatedly from any $\pi \in \Pi$ converges to a policy $\pi^*$ such that $Q^{\pi^*}(\mathbf{s}_t, \mathbf{a}_t) \geq Q^\pi(\mathbf{s}_t, \mathbf{a}_t)$ for all $\pi \in \Pi$ and $(\mathbf{s}_t, \mathbf{a}_t) \in \mathcal{S} \times \mathcal{S}$.*

*Proof.* See Appendix A.2 □

---

**Algorithm 1:** Diffusion Q-Sampling (DQS)

---

**Initialize:** Initialize Q-function parameters $\theta$, policy parameters $\phi$, target network $\bar{\theta} \leftarrow \theta$,
           replay buffer $\mathcal{D}$

**for** *each iteration* **do**

    // Environment Interaction

    **for** *each environment step* **do**

        Observe state $\mathbf{s}_t$ and sample action $\mathbf{a}_t$ via reverse diffusion using $f_\phi$

        Execute $\mathbf{a}_t$, observe reward $r_t$ and next state $\mathbf{s}_{t+1}$

        Store transition $(\mathbf{s}_t, \mathbf{a}_t, r_t, \mathbf{s}_{t+1})$ in $\mathcal{D}$

    **end**

    // Parameter Updates

    **for** *each gradient step* **do**

        Sample minibatch $\mathcal{B} = \{(\mathbf{s}_t, \mathbf{a}_t, r_t, \mathbf{s}_{t+1})\}$ from $\mathcal{D}$

        // Update Q-function parameters $\theta$

        Compute target Q-values: $\hat{Q}_t = r_t + \gamma Q_{\bar{\theta}}(\mathbf{s}_{t+1}, \mathbf{a}_{t+1}), \quad \mathbf{a}_{t+1} \sim \pi_\phi(\mathbf{s}_{t+1})$

        Update $\theta$ by minimizing: $J(\theta) = \frac{1}{|\mathcal{B}|} \sum_{\mathcal{B}} \left( Q_\theta(\mathbf{s}_t, \mathbf{a}_t) - \hat{Q}_t \right)^2$

        // Update policy parameters $\phi$

        **for** *each $(\mathbf{s}_t, \mathbf{a}_t)$ in $\mathcal{B}$* **do**

            Sample diffusion time $\tau \sim \mathcal{U}[0, 1]$

            Sample noisy action $\mathbf{a}_{t,\tau} \sim \mathcal{N}(\mathbf{a}_t, \sigma_\tau^2 \mathbf{I})$

            Sample $\{\tilde{\mathbf{a}}_t^{(i)}\}_{i=1}^K$, where $\tilde{\mathbf{a}}_t^{(i)} \sim \mathcal{N}(\mathbf{a}_{t,\tau}, \sigma_\tau^2 \mathbf{I})$

            Estimate score: $S_t = \nabla_{\mathbf{a}_{t,\tau}} \log \sum_{i=1}^K \exp \left( Q_\theta(\mathbf{s}_t, \tilde{\mathbf{a}}_t^{(i)}) \right)$

            Update $\phi$ by minimizing: $J(\phi) = \|f_\phi(\mathbf{s}_t, \mathbf{a}_{t,\tau}, \tau) - S_t\|^2$

        **end**

        // Update target network

        Update $\bar{\theta} \leftarrow \eta \theta + (1 - \eta) \bar{\theta}$

    **end**

**end**

---

## 4.2 DIFFUSION Q-SAMPLING

In this section, we propose an off-policy actor-critic algorithm, which we call *Diffusion Q-Sampling* (DQS), based on the above framework. Being an off-policy method means DQS can reuse past interactions with the environment by storing them in a replay buffer $\mathcal{D}$, improving sample efficiency.

Let $Q_\theta$ denote the Q-function and $\pi_\phi$ a parametric policy, where $\theta, \phi$ represent the parameters of a neural network. The Q-function is learned using standard temporal difference learning,

$$J(\theta) = \mathbb{E}_{(\mathbf{s}_t, \mathbf{a}_t) \sim \mathcal{D}} \left[ \left( Q_\theta(\mathbf{s}_t, \mathbf{a}_t) - \hat{Q}(\mathbf{s}_t, \mathbf{a}_t) \right)^2 \right], \tag{8}$$

where

$$\hat{Q}(\mathbf{s}_t, \mathbf{a}_t) = r(\mathbf{s}_t, \mathbf{a}_t) + \gamma \, \mathbb{E}_{\mathbf{s}_{t+1} \sim \mathcal{P}, \mathbf{a}_{t+1} \sim \pi_\phi} \left[ Q_{\bar{\theta}}(\mathbf{s}_{t+1}, \mathbf{a}_{t+1}) \right]$$

The target Q-values, $\hat{Q}$, make use of a target Q-network denoted by $Q_{\bar{\theta}}$, where the parameters $\bar{\theta}$ are usually an exponentially moving average of the Q-network parameters $\theta$. Also, in practice, the expectation over next states $\mathbf{s}_{t+1}$ is estimated using only a single sample.

We parameterize the policy using a diffusion process and use iDEM (Akhound-Sadegh et al., 2024) to sample actions from the target density $\pi(\cdot | \mathbf{s}_t) \propto \exp(Q_\theta(\mathbf{s}_t, \mathbf{a}_t))$.

**Forward process.** Given $(\mathbf{s}_t, \mathbf{a}_t) \in \mathcal{S} \times \mathcal{A}$, we progressively add Gaussian noise to the action following some noise schedule. Let $\mathbf{a}_{t,\tau}$ denote the noisy action at diffusion step $\tau \in [0, 1]$, such that,

$$\mathbf{a}_{t,0} = \mathbf{a}_t; \qquad \mathbf{a}_{t,\tau} \sim \mathcal{N}(\mathbf{a}_t, \sigma_\tau^2 I).$$

We choose a geometric noise schedule $\sigma_\tau = \sigma_{\min} \left( \frac{\sigma_{\max}}{\sigma_{\min}} \right)^\tau$, where $\sigma_{\min}$ and $\sigma_{\max}$ are hyperparameters. We found it sufficient to set $\sigma_{\min} = 10^{-5}$ and $\sigma_{\max} = 1.0$ for all our experiments.

**Reverse process.** Given noisy action samples, we iteratively denoise them using a learned score function to produce a sample from the target action distribution $\pi(\cdot|\mathbf{s}_t) \propto \exp(Q^\pi(\mathbf{s}_t, \cdot))$. We train a neural network, $f_\phi$ to match iDEM's $K$-sample Monte Carlo estimator of the score, defined in Equation (5), by setting the negative Q-function as the energy function. The score function takes as input the noisy action and diffusion time, while also being conditioned on the current state. The loss function is given by:

$$J(\phi) = \mathbb{E}_{\substack{(\mathbf{s}_t, \mathbf{a}_t) \sim \mathcal{D}, \tau \sim U[0,1], \\ \mathbf{a}_{t,\tau} \sim \mathcal{N}(\mathbf{a}_t, \sigma_\tau^2 I)}} \left[ \left\| f_\phi(\mathbf{s}_t, \mathbf{a}_{t,\tau}, \tau) - \nabla \log \sum_{i=1}^{K} \exp(Q_\theta(\mathbf{s}_t, \tilde{\mathbf{a}}_t^{(i)})) \right\|^2 \right], \quad (9)$$

where $\tilde{\mathbf{a}}_t^{(i)} \sim \mathcal{N}(\mathbf{a}_{t,\tau}, \sigma_\tau^2 I)$.

Summarizing, to sample an action $\mathbf{a}_t$ for the current state $\mathbf{s}_t$ such that $\pi_\phi(\mathbf{a}_t|\mathbf{s}_t) \propto \exp(Q^{\pi_\phi}(\mathbf{s}_t, \mathbf{a}_t))$, we first sample noise from the prior (corresponding to diffusion time $\tau = 1$) $\mathbf{a}_{t,1} \sim \mathcal{N}(0, \sigma_1^2)$. We then use Equation (3) in the VE setting (i.e. $\alpha(\tau) = 0$) by using the trained score function $f_\phi$ in place of $\nabla \log p_\tau$ to iteratively denoise samples produce the action sample $\mathbf{a}_t$. The full algorithm is presented in Algorithm 1.

**Temperature.** We can incorporate the temperature parameter $T$ from Equation (6) within our framework by simply scaling the Q-function in Equation (9) and regressing to the estimated score of the temperature-scaled Boltzmann distribution. To enable the score network to model this temperature scaling accurately, we additionally condition $f_\phi$ on the current temperature. Generally, the temperature is set to a high value initially, and is annealed over time such that at $t \to \infty$, we have $T \to 0$. In practice, the temperature is annealed to a sufficiently small value for large time steps. This ensures that the policy explores initially and as it collects more information about the environment, starts exploiting more and more as time passes.

## 5 EXPERIMENTS

We perform experiments to answer the following major questions:

- Can DQS learn multimodal behaviors, i.e., learn multiple ways to solve a task?
- Does DQS offer better exploration-exploitation in continuous control tasks?

**Baselines.** We test our method against two relevant baseline methods: (1) Soft Actor-Critic (SAC) (Haarnoja et al., 2018a), a maximum entropy RL method that is widely used for continuous state-action spaces, and; (2) Q-Score Matching (QSM) (Psenka et al., 2023), a recent diffusion-based approach that trains a score function to match the gradient of the Q-function and uses this score function to sample actions.

All methods were trained with $250k$ environment interactions and one network update per environment step. For a fair comparison, all policy/score networks are MLPs with two hidden layers of dimension 256 each, and the learning rate for all networks is $3 \times 10^{-4}$. We apply the double Q-learning trick in our implementation, a commonly used technique, where two Q-networks are trained independently and their minimum value is used for policy evaluation to avoid overestimation bias.

### 5.1 GOAL REACHING MAZE

We use a custom maze environment to evaluate the ability of our method to reach multiple goals. The agent is tasked with manipulating a ball to reach some unknown goal position in the maze. The state consists of the ball's $(x, y)$ position and the velocity vector. The action is the force vector applied to the ball. The initial state of the ball is at the center of the maze, with some noise added for variability. We define two potential goal states for the ball - the top left and the bottom right corners

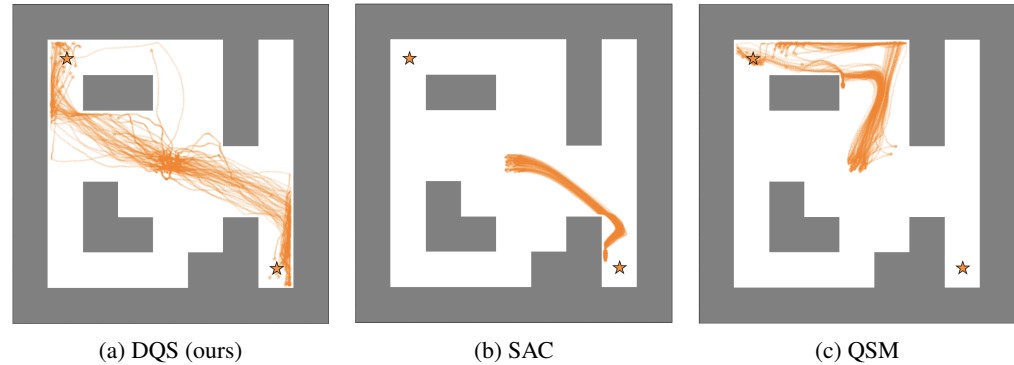

(a) DQS (ours)          (b) SAC          (c) QSM

Figure 2: Trajectories for 100 evaluation episodes after $250k$ training steps.

respectively. The negative Euclidean distance between the desired goal and the achieved state gives the reward function.

For DQS, we used temperature annealing with an initial temperature of $T = 10$, which is decayed exponentially with the number of training steps to a value of $T = 1$ after $250k$ steps. SAC uses automatic temperature tuning (Haarnoja et al., 2018b), where the entropy co-efficient is automatically tuned using gradient descent to maintain the desired level of entropy.

Figure 2 plots the trajectories of the ball over $100$ evaluation episodes after $250k$ training steps. As seen in Figure 2a, DQS learns to reach both goals, owing to the proposed sampling approach which can effectively capture multimodal behavior. Moreover, it discovers both paths to reach the top left goal. In contrast, both SAC (Figure 2b) and QSM (Figure 2c) can only reach one of the goals. Since SAC models the policy using a Gaussian, there is little variability between different trajectories. QSM produces slightly more varied behavior, since it also uses a diffusion model to sample actions, but ultimately fails to learn distinct behaviors.

## 5.2 DEEPMIND CONTROL SUITE

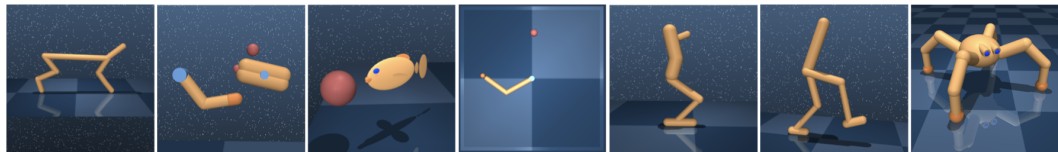

Figure 3: Domains from DeepMind Control Suite considered in our experiments - cheetah, finger, fish, reacher, hopper, quadruped and walker.

We evaluate the performance of DQS on several continuous control tasks via the DeepMind Control Suite. We choose eight tasks from different domains to cover tasks of varying complexity and dynamics. These tasks typically involve controlling the torques applied at the joints of robots to reach a specific configuration or location, or for locomotion.

Since we are interested in evaluating the ability of DQS to balance exploration and exploitation, we focus on data efficiency by limiting the number of environment interactions to $250k$ steps. Figure 4 shows the performance of various methods on these different tasks. On most tasks, DQS performs on par or outperforms the baseline methods. In particular, on five out of the eight tasks considered (cheetah-run, finger-spin, fish-swim, reacher-hard and walker-run) DQS reaches higher reward much faster than competing methods, demonstrating improved exploration.

We use a single fixed temperature of $T = 0.05$ across tasks. Note that as mentioned above, SAC uses automatic temperature tuning which allows it to influence the policy entropy over the course of training. The performance of DQS may be further improved by fine-tuning the temperature schedule on each individual task.

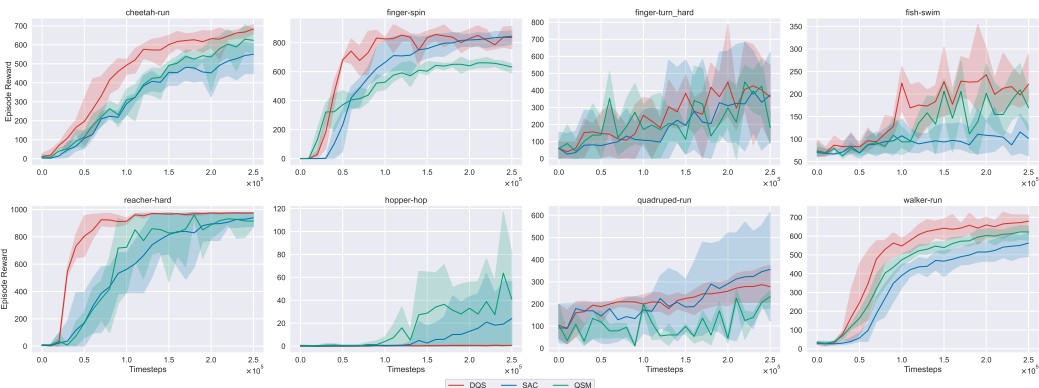

Figure 4: Experimental results on 8 tasks from different domains from the DeepMind Control Suite. Each result is averaged over 10 seeds, with the shaded regions showing minimum and maximum values.

## 6 DISCUSSION

In this work, we showcase the benefits of using energy-based policies as an expressive class of policies for deep reinforcement learning. Such policies arise in different RL frameworks, but their application has been limited in continuous action spaces owing to the difficulty of sampling in this setting. We alleviate this problem using a diffusion-based sampling algorithm. We show that the process of alternating between updating the Q-function of the current policy and optimizing the policy to sample from the Boltzmann distribution of this Q-function has a unique fixed point which is also the optimal policy. The proposed actor-critic algorithm, Diffusion Q-Sampling (DQS,) can sample multimodal behaviors and effectively manage exploration and exploitation.

While diffusion methods offer high expressivity, they often come with increased computation. This is particularly true in the online RL setting, where using a diffusion policy means that each environment step requires multiple function evaluations to sample from the diffusion model. There is a growing body of work on efficient SDE samplers (Jolicoeur-Martineau et al., 2021), which aim to reduce the number of function evaluations required to obtain diffusion-based samples while maintaining high accuracy. Incorporating such techniques with Boltzmann policies can greatly reduce the computational cost, especially in high-dimensional state-action spaces.

A crucial aspect of energy-based policies is the temperature parameter, which defines the shape of the sampling distribution. Our method enables annealing of the temperature from some starting value to lower values, as is typically done when applying Boltzmann policies in deep RL. However, this temperature schedule has to be manually tuned. Haarnoja et al. (2018b) proposes an automatic temperature tuning method for SAC, which maintains the temperature so that the entropy of the current policy is close to some target entropy. While such an approach could be applied to DQS in principle, it is computationally expensive to compute the likelihoods of samples under a diffusion model.

Finally, as we argued in the introduction, Boltzmann policies based on their own value function are attractive choices for pre-training of RL agents for later fine-tuning and multi-task settings. We hope to investigate this exciting potential in the future.

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

# A PROOFS

## A.1 LEMMA 1 (POLICY IMPROVEMENT)

**Lemma 1.** *Let $\pi_{old} \in \Pi$ and $\pi_{new}$ be the policy from Equation (6) with respect to $Q^{\pi_{old}}$. Then $Q^{\pi_{new}}(\mathbf{s}_t, \mathbf{a}_t) \geq Q^{\pi_{old}}(\mathbf{s}_t, \mathbf{a}_t)$ for all $\mathbf{s}_t, \mathbf{a}_t \in \mathcal{S}, \mathcal{A}$.*

*Proof.* Let $Q^{\pi_{old}}$ be the Q-function corresponding to $\pi_{old}$. Since $\pi_{new}$ is defined using Equation (6) corresponding to $Q^{\pi_{old}}$, it must follow that,

$$D_{\mathrm{KL}}\left(\pi_{\mathrm{new}}(\cdot|s_t)||\exp(Q^{\pi_{old}}(s_t,\cdot) - \log Z^{\pi_{old}}(s_t)\right) \leq D_{\mathrm{KL}}\left(\pi_{\mathrm{old}}(\cdot|s_t)||\exp(Q^{\pi_{old}}(s_t,\cdot) - \log Z^{\pi_{old}}(s_t)\right),$$

where $D_{\mathrm{KL}}(p||q)$ denotes the Kullback-Leibler divergence between two distributions $p$ and $q$, and $Z^\pi$ denotes the partition function $Z^\pi(s) = \int_a Q^\pi(s,a)da$.

Since the partition function depends only on the state, the inequality reduces to,

$$\mathbb{E}_{a_t \sim \pi_{\mathrm{new}}}\left[\log \pi_{\mathrm{new}}(a_t|s_t) - Q^{\pi_{old}}(s_t, a_t) + \log Z^{\pi_{old}}(s_t)\right]$$
$$\leq \mathbb{E}_{a_t \sim \pi_{\mathrm{old}}}\left[\log \pi_{\mathrm{old}}(a_t|s_t) - Q^{\pi_{old}}(s_t, a_t) + \log Z^{\pi_{old}}(s_t)\right]$$
$$\Rightarrow \mathbb{E}_{a_t \sim \pi_{\mathrm{old}}}\left[Q^{\pi_{old}}(s_t, a_t)\right] \leq \mathbb{E}_{a_t \sim \pi_{\mathrm{new}}}\left[Q^{\pi_{old}}(s_t, a_t) - \log \pi_{\mathrm{new}}(a_t|s_t) + \log \pi_{\mathrm{old}}(a_t|s_t)\right]. \quad (10)$$

Now, consider the non-negativity of the KL divergence between $\pi_{\mathrm{new}}$ and $\pi_{\mathrm{old}}$,

$$D_{\mathrm{KL}}\left(\pi_{\mathrm{new}}(\cdot|s_t)||\pi_{\mathrm{old}}(\cdot|s_t)\right) \geq 0$$
$$\Rightarrow \mathbb{E}_{a_t \sim \pi_{\mathrm{new}}}\left[\log \pi_{\mathrm{new}}(a_t|s_t) - \log \pi_{\mathrm{old}}(a_t|s_t)\right] \geq 0 \quad (11)$$

Combining Equation (12) and Equation (11), we get,

$$\mathbb{E}_{a_t \sim \pi_{\mathrm{old}}}\left[Q^{\pi_{old}}(s_t, a_t)\right] \leq \mathbb{E}_{a_t \sim \pi_{\mathrm{new}}}\left[Q^{\pi_{old}}(s_t, a_t)\right] \quad (12)$$

Now, consider the Bellman equation,

$$Q^{\pi_{old}}(s_t, a_t) = r(s_t, a_t) + \gamma \mathbb{E}_{s_{t+1} \sim \mathcal{P}}\left[\mathbb{E}_{a_{t+1} \sim \pi_{\mathrm{old}}}\left[Q^{\pi_{old}}(s_{t+1}, a_{t+1})\right]\right]$$
$$\leq r(s_t, a_t) + \gamma \mathbb{E}_{s_{t+1} \sim \mathcal{P}}\left[\mathbb{E}_{a_{t+1} \sim \pi_{\mathrm{new}}}\left[Q^{\pi_{old}}(s_{t+1}, a_{t+1})\right]\right] \quad \text{(from Equation (12))}$$
$$\leq r(s_t, a_t) + \gamma \mathbb{E}_{s_{t+1} \sim \mathcal{P}}\left[\mathbb{E}_{a_{t+1} \sim \pi_{\mathrm{new}}}\left[r(s_{t+1}, a_{t+1}) + \gamma \mathbb{E}[\ldots]\right]\right]$$
$$\vdots$$
$$\leq Q^{\pi_{new}}(s_t, a_t), \quad (13)$$

where we repeatedly expand $Q^{\pi_{old}}$ in the RHS by applying the Bellman equation. The RHS converges to $Q^{\pi_{new}}$ following the fact that the Bellman operator is a contraction. $\square$

## A.2 THEOREM 1 (POLICY ITERATION)

**Theorem 1.** *Alternating between policy evaluation and policy improvement repeatedly from any $\pi \in \Pi$ converges to a policy $\pi^*$ such that $Q^{\pi^*}(\mathbf{s}_t, \mathbf{a}_t) \geq Q^\pi(\mathbf{s}_t, \mathbf{a}_t)$ for all $\pi \in \Pi$ and $(\mathbf{s}_t, \mathbf{a}_t) \in \mathcal{S} \times \mathcal{S}$.*

*Proof.* Let $\pi_i$ be the policy at iteration $i$. By Lemma 1, $Q^{\pi_i}$ is monotonically increasing. Since the reward is bounded, $Q^\pi$ is also bounded for all $\pi \in \Pi$. This sequence converges to some policy $\pi^*$. At convergence, we must have for all $\pi in \Pi$,

$$D_{\mathrm{KL}}\left(\pi^*(\cdot|s_t)||\exp(Q^{\pi^*}(s_t,\cdot) - \log Z^{\pi^*}(s_t))\right) \leq D_{\mathrm{KL}}\left(\pi(\cdot|s_t)||\exp(Q^{\pi^*}(s_t,\cdot) - \log Z^{\pi^*}(s_t))\right).$$

Using the same argument as in Lemma 1, we have $Q^{\pi^*}(\mathbf{s}_t, \mathbf{a}_t) \geq Q^\pi(\mathbf{s}_t, \mathbf{a}_t)$ for all $(\mathbf{s}_t, \mathbf{a}_t) \in \mathcal{S} \times \mathcal{A}$, that is, the Q-value of any policy in $\Pi$ is lower than that of $\pi^*$. Hence, $\pi^*$ is optimal. $\square$

## B  IMPLEMENTATION DETAILS

The score function is parameterized as an MLP with two hidden layers of 256 units each with the ReLU activation function, except for the final layer. The MLP has skip connections as is typical for denoising score functions. The input to the policy comprises the state, noised action, the diffusion time step, and the temperature. The diffusion time step and the temperature are encoded using sinusoidal positional embeddings of 256 dimensions. The action is sampled following Equation (3) and the $\tanh(\cdot)$ function is applied to the sampled action followed by multiplication with the maximum value of the action space to ensure the value is within the correct range. The Q-network is also an MLP with two hidden layers of 256 units each with the ReLU activation function, except for the final layer. We use two Q-networks for the double Q-learning technique, and take the minimum of the two values.

The score function and the Q-network are trained for $250k$ environment steps with one mini-batch update per environment step. Optimization is performed using the Adam optimizer (Kingma, 2014) with a learning rate of $3 \times 10^{-4}$ and a batch size of 256.

Table 1: Hyperparameters.

| Parameter | Value |
|---|---|
| Number of hidden layers | 2 |
| Number of hidden units per layer | 256 |
| Sinusoidal embedding dimension | 256 |
| Activation function | ReLU |
| Optimizer | Adam |
| Learning rate | $3 \cdot 10^{-4}$ |
| Number of samples ber minibatch | 256 |
| Replay buffer size | 250000 |
| Discount factor | 0.99 |
| Gradient updates per step | 1 |
| Target smoothing co-efficient | 0.005 |
| Target update period | 1 |
| Seed training steps | $10^4$ |
| $\sigma_{\min}$ | 0.00001 |
| $\sigma_{\max}$ | 1 |
| Number of Monte Carlo samples | 1000 |
| Number of integration steps | 1000 |

