# OpenReview forum: "Sampling from Energy-based Policies using Diffusion"
_ICLR.cc/2025/Conference — ICLR 2025 Conference Withdrawn Submission_

### Official Review · Reviewer_hBVq · 2024-10-24

**Soundness:** 1
**Presentation:** 3
**Contribution:** 2
**Rating:** 3
**Confidence:** 4

**Summary:**

The authors introduce a new algorithm for continuous RL environments, Diffusion Q-Sampling (DQS).
DQS makes use of an existing method, iterated Denoising Energy matching (iDEM).
The key idea is to use iDEM to learn a score function which can be used in a reverse diffusion process to sample actions.
The score function is trained such that the reverse diffusion process approximately samples from a Boltzmann distribution with respect to the Q-function of the current policy.

The authors give two theoretical results, corresponding to policy improvement and policy iteration respectively, to justify their choice of training rule for the action-value function and the diffusion model.

The authors then give experimental results for their method, DQS.
In the first set of results, they compare DQS to SAC (soft actor-critic) and QSM (Q-score matching) in terms of the diversity of behaviors learned. They demonstrate that, in a goal reaching maze environment, DQS can successfully learn a diverse set of solutions, while SAC and QSM learn a more concentrated set of solutions.
In the second set of results, they compare DQS to SAC and QSM on 8 tasks from the DeepMind control suite. They demonstrate that on many of these tasks, QSM dominates the other methods.

**Strengths:**

Originality - The application of iDEM is (to this reviewer's knowledge) novel; although other methods seek to use diffusion model policies, they typically use other methods for fitting the diffusion model. The application of iDEM is novel.

Quality - The empirical results given are strong. The first set of results demonstrates well that DQS can indeed learn a policy which has support on multiple different solution types for problems. The second set of results shows that DQS can learn well, and outperform baseline methods in terms of sample efficiency.

Clarity - In general, the authors writing is clear. The method is well-explained, and seems reproducible.

Significance - The authors propose an effective new algorithm for continuous control. This algorithm seems particularly useful for the setting where compute is not a bottleneck, and multimodal policies are explicitly desired.

**Weaknesses:**

046 - The authors give methods of policy representations in the continuous setting. I would suggest that they mention SQL, which allows for the training of expressive policies which come from neither noise injection nor parametric family. These are trained via Stein-variational gradient descent.

071 - The claim is made that "[Diffusion models] have been extensively applied to solve sequential decision-making tasks, especially in offline settings where they can model multimodal datasets from suboptimal policies or diverse human demonstrations." No citations are given for these techniques - please include citations to the literature to which you are referring.

191 - I would encourage the authors to say more about the role of the reverse SDE (3) in generation. Specifically, please be clear about how (3) is used to generate samples, rather than assuming this knowledge on the part of the reader.

205 - Missing tildes over the x's in the expectations in Eq. (4).

210 - Subscript below the S in equation (5) should be a capital K.

260 - Lemma 1 is false, and its proof is invalid. Lemma 1 states that, for any action-value function, the policy which is Boltzmann with respect to that action-value function has a dominating action-value function. This statement is incorrect, and obviously so. Let $\pi^*$ be the optimal policy, with action-value function $Q^*$. Then we know that $Q^*$ satisfies the Bellman optimality operator, $T^* Q(s,a) = r + \gamma \mathbb{E}[ \max_{a'} Q(s',a') | s, a]$, where the expectation is taken over next-states $s'$ conditional on state-action pair $s,a$. If Lemma 1 were true, it would mean that the Boltzmann policy $\pi_B$ with respect to $Q^*$ has an action-value function which dominates $Q^*$. But note that $T^* Q^*(s,a) \geq T^{\pi_B} Q^*(s,a)$, which can be seen by expanding definitions and using the fact that the maximum over $a'$ dominates any expectation with respect to $a'$, except if that expectation only places mass on the argmax actions. From this it also follows that this inequality is strict somewhere provided $Q^*$ is non-uniform somewhere. But since $T^* Q^* = Q^*$, it follows that $Q^* \geq T^{\pi_B} Q^*$. But monotonicity of the Bellman operator, it follows that, for all $n$, $Q^* \geq [T^{\pi_B}]^n Q^*$. Taking limits as $n \to \infty$, we obtain that, $Q^* \geq Q^{\pi_B}$, with strict inequality somewhere provided $Q^*$ is not flat. This contradicts the stated result.

We now turn to the proof given in A.1, and examine the error of reasoning. In the first two lines of  (10), the expectation of $\log(\pi_{new})$ is taken with respect to $\pi_{new}$, and the expectation of $\log(\pi_{old})$ is taken with respect to $\pi_{old}$. However, in the third line of (10), the expectation of both terms is taken with respect to $\pi_{new}$. This allows the authors to express this term as a KL-divergence, a step critical to their proof. However, the term should instead be a difference of entropies, which in general is not non-negative (as the KL-divergence is).

265 - The proof of Theorem 1 is invalid. The proof relies heavily on the same argument as in Lemma 1, which is faulty.

In general, it seems like the authors fail to appreciate that results from the entropy regularised setting and the classical setting cannot be freely interchanged. The optimal policy is Boltzmann only if an entropy regularisation term is included in the Bellman backup, (7). When there is no such entropy term in the backup, the optimal policy will simply be the classical optimal policy, which in general is deterministic (or has support only on argmax actions). Similarly, the Boltzmann improvement map only gives improvement with entropy regularisation. Otherwise it can result in a strictly worse policy, as explained above.

I would suggest that the authors either cut their theoretical results entirely, or think about replacing the Bellman backup in (7) with the entropy regularized backup - however this would result in a substantial change to the algorithm, which may be too late at this stage.

**Questions:**

074, 122 - The claim is made twice that for Q-Score Matching (Psenka et al. 2023), "the exact form of the policy is unspecified and it is unknown what distribution the diffusion models sample from". But this is equally true for your method. Both DQS and QSM train score functions which are used to generate samples of the policy. And both DQS and QSM aim for these score functions to allow for sampling from the Boltzmann distribution with respect to the current action-value function. So it is unclear what this comment is meant to mean, or what advantage you are supposing DQS has over QSM. Can the authors please clarify this?

461 - You mention that diffusion based policies have an increased runtime compute requirement compared to parametric policy methods. Can you give an indication of the ratio of runtime for your method vs. SAC? Are there experiments you can run which demonstrate that DQS outperforms SAC when normalised for compute time?

Will a codebase be made available to accompany the paper?

---

### Official Review · Reviewer_fgmh · 2024-10-24

**Soundness:** 2
**Presentation:** 2
**Contribution:** 2
**Rating:** 3
**Confidence:** 4

**Summary:**

This paper proposes a diffusion-based sampling method that uses a negative Q-function as an energy function for sampling, thus allowing for more expressive policy representations. Based on this approach, an actor-critic method called **Diffusion Q-Sampling (DQS)** is proposed that enables stable learning in diverse environments. Experiments show that the method enhances exploration in continuous control tasks and effectively captures multimodal behaviours, overcoming key limitations of existing methods. However, the core sampling method used in this paper is iDEM leading to a lack of innovation, and the experimental results are insufficient, the multimodal experiments may be problematic (results of the Q-score method), and the baseline algorithm is too few and too simple.

**Strengths:**

- This article proposes sampling with a diffusion strategy obeying a Boltzmann distribution to balance exploration and exploitation, focusing on a very cutting-edge area;
- This paper does a multimodal experiment to show that DQS has some multimodality, a point that may be of interest to the RL community;
- The writing of the paper is easy to follow.

**Weaknesses:**

- The related work is not presented carefully enough, and some are only cited. In particular, the related work under Online diffusion is particularly scarce, and each needs the author to summarise their approach, and where the flaws lie. In addition, **diffusion & online RL** related work also need you to expand, I found a recent paper accepted in NeurIPS24 is also under this setting Diffusion Actor-Critic with Entropy Regulator (Wang et al.).

- You mention that the Q-score method does not have an exact distribution, but isn't Eq. (21) of the original paper a Boltzmann distribution? Is the representation in your paper not quite correct.  It's better to clarify your statement about the Q-score method and explain how it relates to Eq. (21) in the original paper.

- The two proofs in 4.1 about policy improvement and policy iteration do not depend on the diffusion model, this is essentially a mathematical proof of a policy obeying a Boltzmann distribution. May I ask what is the essential difference between your proofs and the one in the Soft Actor-Critic Algorithms and Applications (Haarnoja et al.) paper?

- With the experiments in 5.1, I remain sceptical about the results of QSM. I think with the addition of some tricks to fully learn the bias of Q with respect to a, the QSM can get the same results as you did (e.g., do some random sampling to update the bias of Q with respect to a to get it to school in full action space).

- 5.2 There is too little BASELINE for experimental comparisons. To prove your excellent performance, add Proximal Policy Optimisation Algorithms (Schulman et al.), Diffusion Actor-Critic with Entropy Regulator (Wang et al.), Policy Representation via Diffusion Probability Model for Reinforcement Learning (Yang et al.). At least a few difficult scenarios are tested on MuJoCo environment (Humanoid, Ant) and compared with the above algorithms.

### Minor note.
- Please label all formulas in PRIMARY with the serial number, then you look at the expression for the Q function, where does the discount factor go?

- Equation (4) has incorrect parentheses.

**Questions:**

See suggestions and questions in the Weaknesses section.

---

### Official Review · Reviewer_auFv · 2024-11-02

**Soundness:** 2
**Presentation:** 2
**Contribution:** 2
**Rating:** 3
**Confidence:** 3

**Summary:**

This paper proposes a novel framework for sequential decision-making using diffusion models for
sampling from energy-based policies and a new actor-critic algorithm for training diffusion
policies based on that framework.This algorithm improves the high-cost issue of sampling from
continuous action spaces in traditional maximum entropy reinforcement learning methods. It has
been validated in the authors' custom maze navigation and DeepMind Control Suite tasks.

**Strengths:**

Proposing a novel Boltzmann policy iteration which is more efficiency and still bound to recover the optical policy

**Weaknesses:**

Lack of novelty：Simply integrating Diffusion into the traditional SAC which lacks innovation.

Benchmark in a custom environment lacks persuasiveness and the test is not quantified to data.

**Questions:**

How does the method compare to recent Diffusion RL algorithms that outperform QSM,such as

1.Diffusion-based Reinforcement Learning via Q-weighted Variational Policy Optimization,https://arxiv.org/abs/2405.16173

2.Policy Representation via Diffusion Probability Model for Reinforcement Learning,https://arxiv.org/abs/2305.13122

Can the algorithm demonstrate its advantages in a broader range of test environments, rather
than just in a custom maze?

Can the experiments be quantified into numbers or tables rather than presenting the results
using abstract images?

---

### Official Review · Reviewer_zuam · 2024-11-06

**Soundness:** 3
**Presentation:** 3
**Contribution:** 3
**Rating:** 6
**Confidence:** 3

**Summary:**

The authors have developed a new actor-critic algorithm called Diffusion Q-Sampling (DQS), which uses a diffusion-based model to sample from energy-based policies in actor-critic framework. The goal is to address current limitation of capturing complexity of multimodal action distributions in continuous action spaces. This novel algorithm is shown to be very effective for learning multimodal behaviors and improved sample efficiency.

**Strengths:**

1. The novel approach is able to learn multimodal actions which is valuable especial when multiple optimal trajectory exists.
2. By explicitly sampling from the Boltzmann distribution of the Q function, DQS is shown better abilities for balancing exploration and exploitation.
3. Through experiments on maze tasks and Deepmind control suites benchmarks, results have confirmed the advantages of DQS.

**Weaknesses:**

1. As pointed out by the authors, temperature of DQS needs to be manually tuned unlike SAC as it would be computationally very expensive to compute the likelihoods under diffusion model.
2. No ablation study. Maybe beneficial to have some ablation studies, for example, how sensitive DQS is to different temperature values, K (number of monte carlo samples and how is it relates to computation cost)? or isolate the contribution of techniques introduced, etc.

**Questions:**

For benchmark environments where DQS does not show clear advantages, is there any analysis/explanation?
I think ablation study would be useful. Any justifications why u choose not to have ablations?
I believe DQS is sample efficient as they perform better quicker at the early stage of training for some of the environments, I'm curious how would DQS perform at the late stage of training? Have you ever run the algorithm for longer training iterations (for example, 1m iterations)?

---

### Note · Authors · 2024-11-13

**Comment:**

We thank the reviewers for their feedback, particularly reviewer hBVq for the detailed comments. We will incorporate the suggestions and revise our work.

**Withdrawal Confirmation:**

I have read and agree with the venue's withdrawal policy on behalf of myself and my co-authors.